# Information Constraints on Auto-Encoding Variational Bayes

**Romain Lopez**[1], **Jeffrey Regier**[1], **Michael I. Jordan**[1,2], **and Nir Yosef**[1,3,4]
{romain_lopez, regier, niryosef}@berkeley.edu
jordan@cs.berkeley.edu

[1]Department of Electrical Engineering and Computer Sciences, University of California, Berkeley
[2]Department of Statistics, University of California, Berkeley
[3]Ragon Institute of MGH, MIT and Harvard
[4]Chan-Zuckerberg Biohub

## Abstract

Parameterizing the approximate posterior of a generative model with neural networks has become a common theme in recent machine learning research. While providing appealing flexibility, this approach makes it difficult to impose or assess structural constraints such as conditional independence. We propose a framework for learning representations that relies on auto-encoding variational Bayes, in which the search space is constrained via kernel-based measures of independence. In particular, our method employs the $d$-variable Hilbert-Schmidt Independence Criterion (dHSIC) to enforce independence between the latent representations and arbitrary nuisance factors. We show how this method can be applied to a range of problems, including problems that involve learning invariant and conditionally independent representations. We also present a full-fledged application to single-cell RNA sequencing (scRNA-seq). In this setting the biological signal is mixed in complex ways with sequencing errors and sampling effects. We show that our method outperforms the state-of-the-art approach in this domain.

## 1 Introduction

Since the introduction of variational auto-encoders (VAEs) [1], graphical models whose conditional distribution are specified by deep neural networks have become commonplace. For problems where all that matters is the goodness-of-fit (e.g., marginal log probability of the data), there is little reason to constrain the flexibility/expressiveness of these networks other than possible considerations of overfitting. In other problems, however, some latent representations may be preferable to others—for example, for reasons of interpretability or modularity. Traditionally, such constraints on latent representations have been expressed in the graphical model setting via conditional independence assumptions. But these assumptions are relatively rigid, and with the advent of highly flexible conditional distributions, it has become important to find ways to constrain latent representations that go beyond the rigid conditional independence structures of classical graphical models.

In this paper, we propose a new method for restricting the search space to latent representations with desired independence properties. As in [1], we approximate the posterior for each observation $X$ with an encoder network that parameterizes $q_\phi(Z \mid X)$. Restricting this search space amounts to constraining the class of variational distributions that we consider. In particular, we aim to constrain the *aggregated variational posterior* [2]:

$$\hat{q}_\phi(Z) := \mathbb{E}_{p_{\text{data}}(X)}\left[q_\phi(Z \mid X)\right]. \tag{1}$$

Here $p_{\text{data}}(X)$ denotes the empirical distribution. We aim to enforce independence statements of the form $\hat{q}_\phi(Z^i) \perp\!\!\!\perp \hat{q}_\phi(Z^j)$, where $i$ and $j$ are different coordinates of our latent representation.

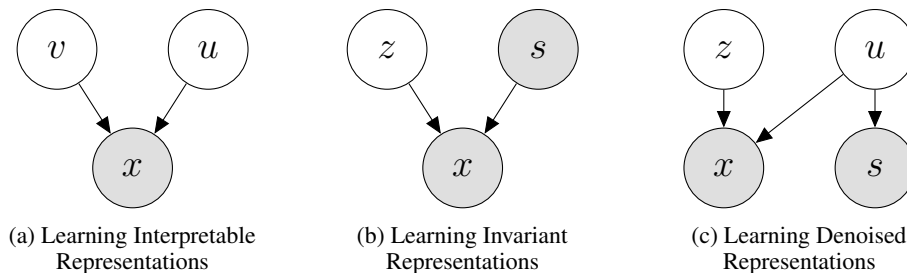

| (a) Learning Interpretable Representations | (b) Learning Invariant Representations | (c) Learning Denoised Representations |

Figure 1: Tasks presented in the paper.

Unfortunately, because $\hat{q}_\phi(Z)$ is a mixture distribution, computing any standard measure of independence is intractable, even in the case of Gaussian terms [3]. In this paper, we circumvent this problem in a novel way. First, we estimate dependency though a kernel-based measure of independence, in particular the Hilbert-Schmidt Information Criterion (HSIC) [4]. Second, by scaling and then subtracting this measure of dependence in the variational lower bound, we get a new variational lower bound on $\log p(X)$. Maximizing it amounts to maximizing the traditional variational lower bound with a penalty for deviating from the desired independence conditions. We refer to this approach as *HSIC-constrained VAE* (HCV).

The remainder of the paper is organized as follows. In Section 2, we provide background on VAEs and the HSIC. In Section 3, we precisely define HCV and provide a theoretical analysis. The next three sections each present an application of HVC—one for each task shown in Figure 1. In Section 4, we consider the problem of learning an interpretable latent representation, and we show that HCV compares favorably to $\beta$-VAE [5] and $\beta$-TCVAE [6]. In Section 5, we consider the problem of learning an invariant representation, showing both that HCV includes the variational fair auto-encoder (VFAE) [7] as a special case, and that it can improve on the VFAE with respect to its own metrics. In Section 6, we denoise single-cell RNA sequencing data with HCV, and show that our method recovers biological signal better than the current state-of-the-art approach.

## 2 Background

In representation learning, we aim to transform a variable $x$ into a *representation vector* $z$ for which a given downstream task can be performed more efficiently, either computationally or statistically. For example, one may learn a low-dimensional representation that is predictive of a particular label $y$, as in supervised dictionary learning [8]. More generally, a hierarchical Bayesian model [9] applied to a dataset yields stochastic representations, namely, the sufficient statistics for the model's posterior distribution. In order to learn representations that respect specific independence statements, we need to bring together two independent lines of research. First, we will present briefly variational auto-encoders and then non-parametric measures of dependence.

### 2.1 Auto-Encoding Variational Bayes (AEVB)

We focus on variational auto-encoders [1] which effectively summarize data for many tasks within a Bayesian inference paradigm [10, 11]. Let $\{X, S\}$ denote the set of observed random variables and $Z$ the set of hidden random variables (we will use the notation $z^i$ to denote the $i$-th random variable in the set $Z$). Then Bayesian inference aims to maximize the likelihood:

$$p_\theta(X \mid S) = \int p_\theta(X \mid Z, S) dp(Z). \tag{2}$$

Because the integral is in general intractable, variational inference finds a distribution $q_\phi(Z \mid X, S)$ that minimizes a lower bound on the data—the evidence lower bound (ELBO):

$$\log p_\theta(X \mid S) \geq \mathbb{E}_{q_\phi(Z|X,S)} \log p_\theta(X \mid Z, S) - D_{KL}((q_\phi(Z|X,S) \,\|\, p(Z)) \tag{3}$$

In auto-encoding variational Bayes (AEVB), the variational distribution is parametrized by a neural network. In the case of a variational auto-encoder (VAE), both the generative model and the variational approximation have conditional distributions parametrized with neural networks. The difference

between the data likelihood and the ELBO is the variational gap:

$$D_{KL}(q_\phi(Z \mid X, S) \,\|\, p_\theta(Z \mid X, S)). \tag{4}$$

The original AEVB framework is described in the seminal paper [1] for the case $Z = \{z\}, X = \{x\}, S = \varnothing$. The representation $z$ is optimized to "explain" the data $x$.

AEVB has since been successfully applied and extended. One notable example is the semi-supervised learning case—where $Z = \{z^1, z^2\}$, $X = \{x\}$, $y \in X \cup Z$—which is addressed by the M1 + M2 model [12]. Here, the representation $z_1$ both explains the original data and is predictive of the label $y$. More generally, solving an additional problem is tantamount to adding a node in the underlying graphical model. Finally, the variational distribution can be used to meet different needs: $q_\phi(y \mid x)$ is a classifier and $q_\phi(z^1 \mid x)$ summarizes the data.

When using AEVB, the empirical data distribution $p_{\text{data}}(X, S)$ is transformed into the empirical representation $\hat{q}_\phi(Z) = \mathbb{E}_{p_{\text{data}}(X,S)} q_\phi(Z \mid X, S)$. This mixture is commonly called the aggregated posterior [13] or average encoding distribution [14].

## 2.2 Non-parametric estimates of dependence with kernels

Let $(\Omega, \mathcal{F}, \mathbb{P})$ be a probability space. Let $\mathcal{X}$ (resp. $\mathcal{Y}$) be a separable metric space. Let $u : \Omega \to \mathcal{X}$ (resp. $v : \Omega \to \mathcal{Y}$) be a random variable. Let $k : \mathcal{X} \times \mathcal{X} \to \mathbb{R}$ (resp. $l : \mathcal{Y} \times \mathcal{Y} \to \mathbb{R}$) be a continuous, bounded, positive semi-definite kernel. Let $\mathcal{H}$ (resp. $\mathcal{K}$) be the corresponding reproducing kernel Hilbert space (RKHS) and $\phi : \Omega \to \mathcal{H}$ (resp. $\psi : \Omega \to \mathcal{K}$) the corresponding feature mapping.

Given this setting, one can embed the distribution $P$ of random variable $u$ into a single point $\mu_P$ of the RKHS $\mathcal{H}$ as follows:

$$\mu_P = \int_\Omega \phi(u) P(du). \tag{5}$$

If the kernel $k$ is universal[1], then the mean embedding operator $P \mapsto \mu_P$ is injective [15].

We now introduce a kernel-based estimate of *distance* between two distributions $P$ and $Q$ over the random variable $u$. This approach will be used by one of our baselines for learning invariant representations. Such a distance, defined via the canonical distance between their $\mathcal{H}$-embeddings, is called the maximum mean discrepancy [16] and denoted $\mathrm{MMD}(P, Q)$.

The joint distribution $P(u, v)$ defined over the product space $\mathcal{X} \times \mathcal{Y}$ can be embedded as a point $\mathcal{C}_{uv}$ in the tensor space $\mathcal{H} \otimes \mathcal{K}$. It can also be interpreted as a linear map $\mathcal{H} \to \mathcal{K}$:

$$\forall (f, g) \in \mathcal{H} \times \mathcal{K}, \ \mathbb{E}f(u)g(v) = \langle f(u), \mathcal{C}_{uv} g(v) \rangle_\mathcal{H} = \langle f \otimes g, \mathcal{C}_{uv} \rangle_{\mathcal{H} \otimes \mathcal{K}}. \tag{6}$$

Suppose the kernels $k$ and $l$ are universal. The largest eigenvalue of the linear operator $\mathcal{C}_{uv}$ is zero if and only if the random variables $u$ and $v$ are marginally independent [4]. A measure of dependence can therefore be derived from the Hilbert-Schmidt norm of the cross-covariance operator $\mathcal{C}_{uv}$ called the Hilbert-Schmidt Independence Criterion (HSIC) [17]. Let $(u_i, v_i)_{1 \le i \le n}$ denote a sequence of iid copies of the random variable $(u, v)$. In the case where $\mathcal{X} = \mathbb{R}^p$ and $\mathcal{Y} = \mathbb{R}^q$, the V-statistics in Equation 7 yield a biased empirical estimate [15], which can be computed in $\mathcal{O}(n^2(p + q))$ time. An estimator for HSIC is

$$
\hat{\mathrm{HSIC}}_n(P) = \frac{1}{n^2} \sum_{i,j}^n k(u_i, u_j) l(v_i, v_j) + \frac{1}{n^4} \sum_{i,j,k,l}^n k(u_i, u_j) l(v_k, v_l)
$$
$$
- \frac{2}{n^3} \sum_{i,j,k}^n k(u_i, u_j) l(v_i, v_k). \tag{7}
$$

The $d$HSIC [18, 19] generalizes the HSIC to $d$ variables. We present the $d$HSIC in Appendix A.

## 3 Theory for HSIC-Constrained VAE (HCV)

This paper is concerned with intepretability of representations learned via VAEs. Independence between certain components of the representation can aid in interpretability [6, 20]. First, we will

explain why AEVB might not be suitable for learning representations that satisfy independence statements. Second, we will present a simple diagnostic in the case where the generative model is fixed. Third, we will introduce HSIC-constrained VAEs (HCV): our method to correct approximate posteriors learned via AEVB in order to recover *independent* representations.

## 3.1 Independence and representations: Ideal setting

The goal of learning representation that satisfies certain independence statements can be achieved by adding suitable nodes and edges to the generative distribution graphical model. In particular, marginal independence can be the consequence of an "explaining away" pattern as in Figure 1a for the triplet $\{u, x, v\}$. If we consider the setting of infinite data and an accurate posterior, we find that independence statements in the generative model are respected in the latent representation:

**Proposition 1.** *Let us apply AEVB to a model $p_\theta(X, Z \mid S)$ with independence statement $\mathcal{I}$ (e.g., $z^i \perp\!\!\!\perp z^j$ for some $(i, j)$). If the variational gap $\mathbb{E}_{p_{data}(X,S)} D_{KL}(q_\phi(Z \mid X, S) \parallel p_\theta(Z \mid X, S))$ is zero, then under infinite data the representation $\hat{q}_\phi(Z)$ satisfies statement $\mathcal{I}$.*

The proof appears in Appendix B. In practice we may be far from the idealized infinite setting if $(X, S)$ are high-dimensional. Also, AEVB is commonly used with a naive mean field approximation $q_\phi(Z \mid X, S) = \prod_k q_\phi(z^k \mid X, S)$, which could poorly match the real posterior. In the case of a VAE, neural networks are also used to parametrize the conditional distributions of the generative model. This makes it challenging to know whether naive mean field or any specific improvement [11, 21] is appropriate. As a consequence, the aggregated posterior could be quite different from the "exact" aggregated posterior $\mathbb{E}_{p_{data}(X,S)} p_\theta(Z \mid X, S)$. Notably, the independence properties encoded by the generative model $p_\theta(X \mid S)$ will often not be respected by the approximate posterior. This is observed empirically in [7], as well as Section 4 and Section 5 of this work.

## 3.2 A simple diagnostic in the case of posterior approximation

A theoretical analysis explaining why the empirical aggregated posterior presents some misspecified correlation is not straightforward. The main reason is that the learning of the model parameters $\theta$ along with the variational parameters $\phi$ makes diagnosis hard. As a first line of attack, let us consider the case where we approximate the posterior of a fixed model. Consider learning a posterior $q_\phi(Z \mid X, S)$ via naive mean field AEVB. Recent work [22, 14, 13] focuses on decomposing the second term of the ELBO and identifying terms, one of which is the total correlation between hidden variables in the aggregate posterior. This term, in principle, promotes independence. However, the decomposition has numerous interacting terms, which makes exact interpretation difficult. As the generative model is fixed in this setting, optimizing the ELBO is tantamount to minimizing the variational gap, which we propose to decompose as

$$
D_{KL}(q_\phi(Z \mid X, S) \parallel p_\theta(Z \mid X, S)) = \sum_k D_{KL}(q_\phi(z^k \mid X, S) \parallel p_\theta(z^k \mid X, S))
$$
$$
+ \mathbb{E}_{q_\phi(Z \mid X, S)} \log \frac{\prod_k p_\theta(z^k \mid X, S)}{p_\theta(Z \mid X, S)}. \tag{8}
$$

The last term of this equation quantifies the misspecification of the mean-field assumption. The larger it is, the more the coupling between the hidden variables $Z$. Since neural networks are flexible, they can be very successful at optimizing this variational gap but at the price of introducing supplemental correlation between $Z$ in the aggregated posterior. We expect this side effect whenever we use neural networks to learn a misspecified variational approximation.

## 3.3 Correcting the variational posterior

We aim to correct the variational posterior $q_\phi(Z \mid X, S)$ so that it satisfies specific independence statements of the form $\forall (i, j) \in \mathcal{S}, \hat{q}_\phi(z^i) \perp\!\!\!\perp \hat{q}_\phi(z^j)$. As $\hat{q}_\phi(Z)$ is a mixture distribution, any standard measure of independence is intractable based on the conditionals $q_\phi(Z \mid X, S)$, even in the common case of mixture of Gaussian distributions [3]. To address this issue, we propose a novel idea: estimate and minimize the dependency via a non-parametric statistical penalty. Given the AEVB framework, let $\lambda \in \mathbb{R}^+$, $\mathcal{Z}_0 = \{z^{i_1}, .., z^{i_p}\} \subset Z$ and $\mathcal{S}_0 = \{s^{j_1}, .., s^{j_q}\} \subset S$. The HCV framework with

independence constraints on $\mathcal{Z}_0 \cup \mathcal{S}_0$ learns the parameters $\theta, \phi$ from maximizing the ELBO from AEVB penalized by

$$-\lambda d\text{HSIC}(\hat{q}_\phi(z^{i_1}, .., z^{i_p})p_{\text{data}}(s^{j_1}, .., s^{j_q})). \qquad (9)$$

A few comments are in order regarding this penalty. First, the $d$HSIC is positive and therefore our objective function is still a lower bound on the log-likelihood. The bound will be looser but the resulting parameters will yield a more suitable representation. This trade-off is adjustable via the parameter $\lambda$. Second, the $d$HSIC can be estimated with the same samples used for stochastic variational inference (i.e., sampling from the variational distribution) and for minibatch sampling (i.e., subsampling the dataset). Third, the HSIC penalty is based only on the variational parameters—not the parameters of the generative model.

## 4 Case study: Learning interpretable representations

Suppose we want to summarize the data $x$ with two independent components $u$ and $v$, as shown in Figure 1a. The task is especially important for data exploration since independent representations are often more easily interpreted.

A related problem is finding latent factors $(z^1, ..., z^d)$ that correspond to real and interpretable variations in the data. Learning independent representations is then a key step towards learning *disentangled* representations [6, 5, 23, 24]. The $\beta$-VAE [5] proposes further penalizing the $D_{KL}(q_\phi(z \mid x) \| p(z))$ term. It attains significant improvement over state-of-the art methods on real datasets. However, this penalization has been shown to yield poor reconstruction performance [25]. The $\beta$-TCVAE [6] penalized an approximation of the *total correlation* (TC), defined as $D_{KL}(\hat{q}_\phi(z) \| \prod_k \hat{q}_\phi(z^k))$ [26], which is a measure of multivariate mutual independence. However, this quantity does not have a closed-form solution [3] and the $\beta$-TCVAE uses a biased estimator of the TC—a lower bound from Jensen inequality. That bias will be zero only if evaluated on the whole dataset, which is not possible since the estimator has quadratic complexity in the number of samples. However, the bias from the HSIC [17] is of order $\mathcal{O}(1/n)$; it is negligible whenever the batch-size is large enough. HSIC therefore appears to be a more suitable method to enforce independence in the latent space.

To assess the performance of these various approaches to finding independent representations, we consider a linear Gaussian system, for which exact posterior inference is tractable. Let $(n, m, d) \in \mathbb{N}^3$ and $\lambda \in \mathbb{R}^+$. Let $(A, B) \in \mathbb{R}^{d \times n} \times \mathbb{R}^{d \times m}$ be random matrices with iid normal entries. Let $\Sigma \in \mathbb{R}^{d \times d}$ be a random matrix following a Wishart distribution. Consider the following generative model:

$$v \sim \text{Normal}(0, I_n)$$
$$u \sim \text{Normal}(0, I_m) \qquad (10)$$
$$x \mid u, v \sim \text{Normal}(Av + Bu, \lambda I_d + \Sigma).$$

The exact posterior $p(u, v \mid x)$ is tractable via block matrix inversion, as is the marginal $p(x)$, as shown in Appendix C. We apply HCV with $Z = \{u, v\}, X = \{x\}, S = \varnothing, \mathcal{Z}_0 = \{u, v\}, and \mathcal{S}_0 = \varnothing$. This is equivalent to adding to the ELBO the penalty $-\lambda\text{HSIC}(\mathbb{E}_{p_{\text{data}}(x)}q_\phi(u, v \mid x))$. Appendix D describes the stochastic training procedure. We report the trade-off between correlation of the representation and the ELBO for various penalty weights $\lambda$ for each algorithm: $\beta$-VAE [5], $\beta$-TCVAE [6], an unconstrained VAE, and HCV. As correlation measures, we consider the summed Pearson correlation $\sum_{(i,j)} \rho(\hat{q}_\phi(u^i), \hat{q}_\phi(v^j))$ and HSIC.

Results are reported in Figure 2. The VAE baseline (like all the other methods) has an ELBO value worse than the marginal log-likelihood (horizontal bar) since the real posterior is not likely to be in the function class given by naive mean field AEVB. Also, this baseline has a greater dependence in the aggregated posterior $\hat{q}_\phi(u, v)$ than in the exact posterior $\hat{p}(u, v)$ (vertical bar) for the two measures of correlation. Second, while correcting the variational posterior, we want the best trade-off between model fit and independence. HCV attains the highest ELBO values despite having the lowest correlation.

## 5 Case study: Learning invariant representations

We now consider the particular problem of learning representations for the data that is *invariant* to a given nuisance variable. As a particular instance of the graphical model in Figure 1b, we embed an

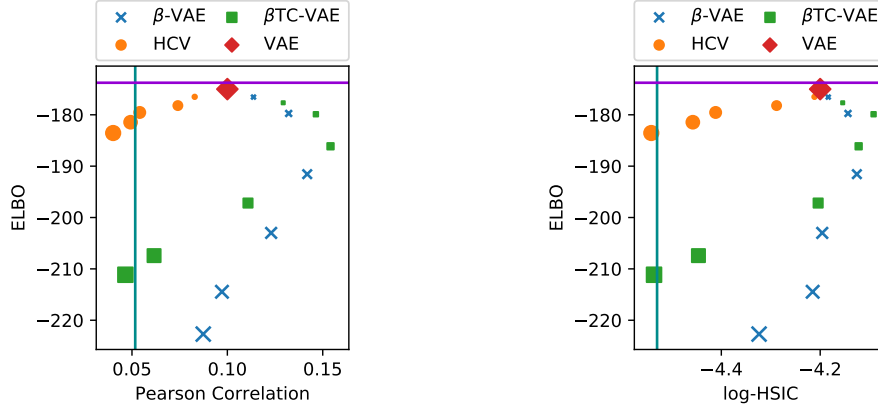

Figure 2: Results for the linear Gaussian system. All results are for a test set. Each dot is averaged across five random seeds. Larger dots indicate greater regularization. The purple line is the log-likelihood under the true posterior. The cyan line is the correlation under the true posterior.

image $x$ into a latent vector $z_1$ whose distribution is independent of the observed lighting condition $s$ while being predictive of the person identity $y$ (Figure 3). The generative model is defined in Figure 3c and the variational distribution decomposes as $q_\phi(z^1, z^2 \mid x, s, y) = q_\phi(z^1 \mid x, s) q_\phi(z^2 \mid z^1, y)$, as in [7].

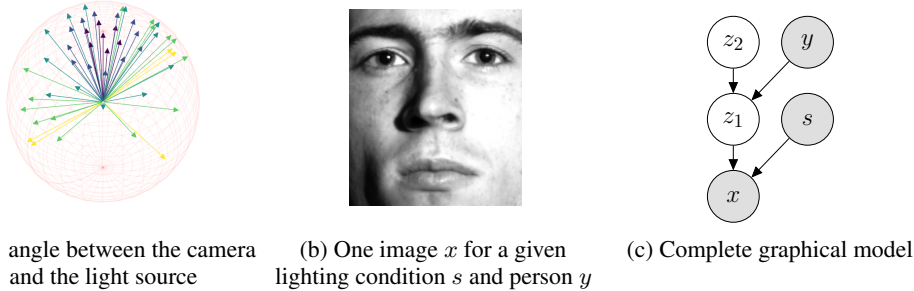

(a) $s$: angle between the camera and the light source

(b) One image $x$ for a given lighting condition $s$ and person $y$

(c) Complete graphical model

Figure 3: Framework for learning invariant representations in the Extended Yale B Face dataset.

This problem has been studied in [7] for binary or categorical $s$. For their experiment with a continuous covariate $s$, they discretize $s$ and use the MMD to match the distributions $\hat{q}_\phi(z^1 \mid s = 0)$ and $\hat{q}_\phi(z^1 \mid s = j)$ for all $j$. Perhaps surprisingly, their penalty turns out to be a special case of our HSIC penalty. (We present a proof of this fact in Appendix D.)

**Proposition 2.** *Let the nuisance factor $s$ be a discrete random variable and let $l$ (the kernel for $\mathcal{K}$) be a Kronecker delta function $\delta : (s, s') \mapsto \mathbb{1}_{s=s'}$. Then, the V-statistic corresponding to $HSIC(\hat{q}_\phi(z^1), p_{data})$ is a weighted sum of the V-statistics of the MMD between the pairs $\hat{q}_\phi(z \mid s = i), \hat{q}_\phi(z \mid s = j)$. The weights are functions of the empirical probabilities for $s$.*

Working with the HSIC rather than an MMD penalty lets us avoid discretizing $s$. We take into account the whole angular range and not simply the direction of the light. We apply HCV with mean-field AEVB, $Z = \{z^1, z^2\}$, $X = \{x, y\}$, $S = \{s\}$, $\mathcal{Z}_0 = \{z^1\}$ and $\mathcal{S}_0 = \{s\}$.

**Dataset** The extended Yale B dataset [27] contains cropped faces [28] of 38 people under 50 lighting conditions. These conditions are unit vectors in $\mathbb{R}^3$ encoding the direction of the light source and can be summarized into five discrete groups (upper right, upper left, lower right, lower left and front). Following [7], we use one image from each group per person (total 190 images) and use the remaining images for testing. The task is to learn a representation of the faces that is good at identifying people but has low correlation with the lighting conditions.

**Experiment**   We repeat the experiments from the paper introducing the variational fair auto-encoder (VFAE) [7], this time comparing the VAE [1] with no covariate $s$, the VFAE [7] with observed lighting direction groups (five groups), and the HCV with the lighting direction vector (a three-dimensional vector). As a supplemental baseline, we also report results for the unconstrained VAEs. As in [7], we report 1) the accuracy for classifying the person based on the variational distribution $q_\phi(y \mid z^1, s)$; 2) the classification accuracy for the lighting group condition (five-way classification) based on a logistic regression and a random forest classifier on a sample from the variational posterior $q_\phi(z^1 \mid z^2, y, s)$ for each datapoint; and 3) the average error for predicting the lighting direction with linear regression and a random forest regressor, trained on a sample from the variational posterior $q_\phi(z^1 \mid z^2, y, s)$. Error is expressed in degrees. $\lambda$ is optimized via grid search as in [7].

We report our results in Table 1. As expected, adding information (either the lightning group or the refined lightning direction) always improves the quality of the classifier $q_\phi(y \mid z^1, s)$. This can be seen by comparing the scores between the vanilla VAE and the unconstrained algorithms. However, by using side information $s$, the unconstrained models yield a representation less suitable because it is more correlated with the nuisance variables. There is therefore a trade-off between correlation to the nuisance and performance. Our proposed method (HCV) shows greater invariance to lighting direction while accurately predicting people's identities.

| | Person identity (Accuracy) | Lighting group (Average classification error) | | Lighting direction (Average error in degree) | |
|---|---|---|---|---|---|
| | | Random Forest Classifier | Logistic Regression | Random Forest Regressor | Linear Regression |
| VAE | 0.72 | 0.26 | 0.11 | 14.07 | 9.40 |
| VFAE* | 0.74 | 0.23 | 0.01 | 13.96 | 8.63 |
| VFAE | 0.69 | 0.51 | **0.42** | 23.59 | 19.89 |
| HCV* | **0.75** | 0.25 | 0.10 | 12.25 | 2.59 |
| HCV | **0.75** | **0.52** | 0.29 | **36.15** | **28.04** |

Table 1: Results on the Extended Yale B dataset. Preprocessing differences likely explain the slight deviation in scores from [7]. Stars (*) the unconstrained version of the algorithm was used.

## 6   Case study: Learning denoised representations

This section presents a case study of denoising datasets in the setting of an important open scientific problem. The task of *denoising* consists of representing experimental observations $x$ and nuisance observations $s$ with two independent signals: biological signal $z$ and technical noise $u$. The difficulty is that $x$ contains both biological signal and noise and is therefore strongly correlated with $s$ (Figure 1c). In particular, we focus on single-cell RNA sequencing (scRNA-seq) data which renders a gene-expression snapshot of an heterogeneous sample of cells. Such data can reveal a cell's type [29, 30], if we can cope with a high level of technical noise [31].

The output of an scRNA-seq experiment is a list of transcripts $(l_m)_{m \in \mathcal{M}}$. Each transcript $l_m$ is an mRNA molecule enriched with a cell-specific barcode and a unique molecule identifier, as in [32]. Cell-specific barcodes enable the biologist to work at single-cell resolution. Unique molecule identifiers (UMIs) are meant to remove some significant part of the technical bias (e.g., amplification bias) and make it possible to obtain an accurate probabilistic model for these datasets [33]. Transcripts are then aligned to a reference genome with tools such as CellRanger [34].

The data from the experiment has two parts. First, there is a gene expression matrix $(X_{ng})_{(n,g) \in \mathcal{N} \times \mathcal{G}}$, where $\mathcal{N}$ designates the set of cells detected in the experiment and $\mathcal{G}$ is the set of genes the transcripts have been aligned with. A particular entry of this matrix indicates the number of times a particular gene has been expressed in a particular cell. Second, we have quality control metrics $(s^i)_{i \in \mathcal{S}}$ (described in Appendix E) which assess the level of errors and corrections in the alignment process. These metrics cannot be described with a generative model as easily as gene expression data but they nonetheless impact a significant number of tasks in the research area [35]. Another significant portion of these metrics focus on the sampling effects (i.e., the discrepancy in the total number of transcripts

captured in each cell) which can be taken into account in a principled way in a graphical model as in [33].

We visualize these datasets $x$ and $s$ with tSNE [36] in Figure 4. Note that $x$ is correlated with $s$, especially within each cell type. A common application for scRNA-seq is discovering cell types, which can be be done without correcting for the alignment errors [37]. A second important application is identifying genes that are more expressed in one cell type than in another—this hypothesis testing problem is called *differential expression* [38, 39]. Not modeling $s$ can induce a dependence on $x$ which hampers hypothesis testing [35].

Most research efforts in scRNA-seq methodology research focus on using generalized linear models and two-way ANOVA [40, 35] to regress out the effects of quality control metrics. However, this paradigm is incompatible with hypothesis testing. A generative approach, however, would allow marginalizing out the effect of these metrics, which is more aligned with Bayesian principles. Our main contribution is to incorporate these alignment errors into our graphical model to provide a better Bayesian testing procedure. We apply HCV with $Z = \{z, u\}, X = \{x, s\}, \mathcal{Z}_0 = \{z, u\}$. By integrating out $u$ while sampling from the variational posterior, $\int q_\phi(x \mid z, u) dp(u)$, we find a Bayes factor that is not subject to noise. (See Appendix F for a complete presentation of the hypothesis testing framework and the graphical model under consideration).

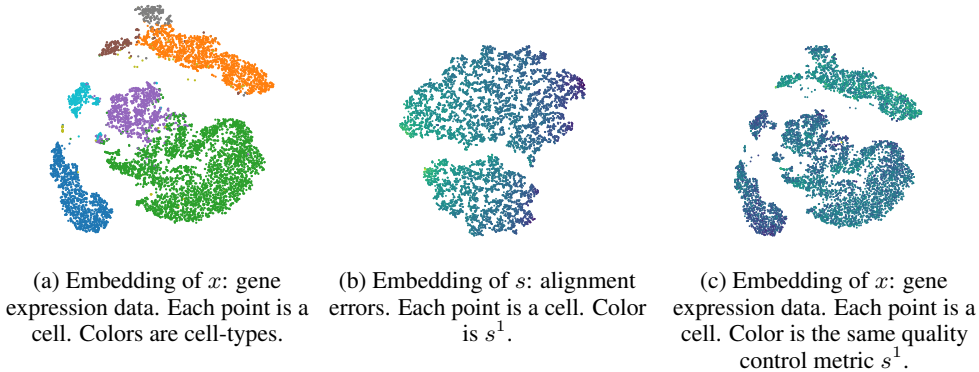

(a) Embedding of $x$: gene expression data. Each point is a cell. Colors are cell-types.

(b) Embedding of $s$: alignment errors. Each point is a cell. Color is $s^1$.

(c) Embedding of $x$: gene expression data. Each point is a cell. Color is the same quality control metric $s^1$.

Figure 4: Raw data from the PBMC dataset. $s^1$ is the proportion of transcripts which confidently mapped to a gene for each cell.

**Dataset** We considered scRNA-seq data from peripheral blood mononuclear cells (PBMCs) from a healthy donor [34]. Our dataset includes 12,039 cells and 3,346 genes, five quality control metrics from CellRanger and cell-type annotations extracted with Seurat [41]. We preprocessed the data as in [33, 35]. Our ground truth for the hypothesis testing, from microarray studies, is a set of genes that are differentially expressed between human B cells and dendritic cells (n=10 in each group [42]).

**Experiment** We compare scVI [33], a state-of-the-art model, with no observed nuisance variables (8 latent dimensions for $z$), and our proposed model with observed quality control metrics. We use five latent dimensions for $z$ and three for $u$. The penalty $\lambda$ is selected through grid search. For each algorithm, we report 1) the coefficient of determination of a linear regression and random forest regressor for the quality metrics predictions based on the latent space, 2) the irreproducible discovery rate (IDR) [43] model between the Bayes factor of the model and the p-values from the micro-array. The mixture weights, reported in [33], are similar between the original scVI and our modification (and therefore higher than other mainstream differential expression procedures) and saturate the number of significant genes in this experiment (~23%). We also report the correlation of the reproducible mixture as a second-order quality metric for our gene rankings.

We report our results in Table 2. First, the proposed method efficiently removes much correlation with the nuisance variables $s$ in the latent space $z$. Second, the proposed method yields a better ranking of the genes when performing Bayesian hypothesis testing. This is shown by a substantially higher correlation coefficient for the IDR, which indicates the obtained ranking better conforms with the micro-array results. Our denoised latent space is therefore extracting information from the data that is less subject to alignment errors and more biologically interpretable.

|      | Irreproducible Discovery Rate | | Quality control metrics (coefficient of determination) | |
| --- | --- | --- | --- | --- |
|      | Mixture weight | Reproducible correlation | Linear Regression | Random Forest Regression |
| scVI | **0.213 ± 0.001** | 0.26 ± 0.07 | 0.195 | 0.129 |
| HCV | **0.217 ± 0.003** | **0.43 ± 0.02** | **0.176** | **0.123** |

Table 2: Results on the PBMCs dataset. IDR results are averaged over twenty initializations.

## 7 Discussion

We have presented a flexible framework for correcting independence properties of aggregated variational posteriors learned via naive mean field AEVB. The correction is performed by penalizing the ELBO with the HSIC—a kernel-based measure of dependency—between samples from the variational posterior.

We illustrated how variational posterior misspecification in AEVB could unwillingly promote dependence in the aggregated posterior. Future work should look at other variational approximations and quantify this dependence.

Penalizing the HSIC as we do for each mini-batch implies that no information is learned about distribution $\hat{q}(Z)$ or $\prod_i \hat{q}(z^i)$ during training. On one hand, this is positive since we do no have to estimate more parameters, especially if the joint estimation would imply a minimax problem as in [23, 13]. One the other hand, that could be harmful if the HSIC could not be estimated with only a mini-batch. Our experiments show this does not happen in a reasonable set of configurations.

Trading a minimax problem for an estimation problem does not come for free. First, there are some computational considerations. The HSIC is computed in quadratic time but linear time estimators of dependence [44] or random features approximations [45] should be used for non-standard batch sizes. For example, to train on the entire extended Yale B dataset, VAE takes two minutes, VFAE takes ten minutes[2], and HCV takes three minutes. Second, the problem of choosing the best kernel is known to be difficult [46]. In the experiments, we rely on standard and efficient choices: a Gaussian kernel with median heuristic for the bandwidth. The bandwidth can be chosen analytically in the case of a Gaussian latent variable and done offline in case of an observed nuisance variable. Third, the general formulation of HCV with the $d$HSIC penalization, as in Equation 9, should be nuanced since the V-statistic relies on a U-statistic of order $2d$. Standard non-asymptotic bounds as in [4] would exhibit a concentration rate of $\mathcal{O}(\sqrt{d/n})$ and therefore not scale well for a large number of variables.

We also applied our HCV framework to scRNA-seq data to remove technical noise. The same graphical model can be readily applied to several other problems in the field. For example, we may wish to remove cell cycles [47] that are biologically variable but typically independent of what biologists want to observe. We hope our approach will empower biological analysis with scalable and flexible tools for data interpretation.

**Acknowledgments**

NY and RL were supported by grant U19 AI090023 from NIH-NIAID.

## Footnotes

[1]A kernel $k$ is universal if $k(x, \cdot)$ is continuous for all $x$ and the RKHS induced by $k$ is dense in $C(\mathcal{X})$. This is true for the Gaussian kernel $(u, u') \mapsto e^{-\gamma \|u - u'\|^2}$ when $\gamma > 0$.

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
