[Supplementary Material · supplementary_materials.pdf]

## A  dHSIC

The definition of the cross-covariance operator can be extended to the case of $d$ random variables, simply by adhering to the formal language of tensor spaces. Let $X = (X^1, ..., X^d)$ be a random vector with each component of dimension $p$. The components $X^1, ..., X^d$ are mutually independent if the joint distribution is equal to the tensor product of the marginal distribution. [18] derives functional formulation, population statistics and V-statistics for the Hilbert-Schmidt norm of the corresponding generalized cross-covariance operator called $d$HSIC. Notably, under the hypothesis that the canonical kernel from the tensor product $\otimes_k \mathcal{H}_k$ is universal, the $d$HSIC is null if and only if the components of the random vector are mutually independent. We write here the retained V-statistics that we will use for the experiments and can compute in time $\mathcal{O}(dn^2 p)$:

$$
\begin{aligned}
\hat{\mathrm{dHSIC}}_n(P) = &\frac{1}{n^2} \sum_{M_2(n)} \prod_{j=1}^d k^j(x^j_{i_1}, x^j_{i_2}) + \frac{1}{n^{2d}} \sum_{M_{2d}(n)} \prod_{j=1}^d k^j(x^j_{i_{2j-1}}, x^j_{i_{2j}}) \\
&- \frac{2}{n^{d+1}} \sum_{M_{d+1}(n)} \prod_{j=1}^d k^j(x^j_{i_1}, x^j_{i_{j+1}})
\end{aligned}
\tag{11}
$$

The question that naturally follows is when is the canonical kernel from the tensor product $\otimes_k \mathcal{H}_k$ universal. [19] showed that the characteristic property of the individual kernels is not enough in general. However, in the case of continuous, bounded and translation invariant kernels this is a sufficient condition. In subsequent work, we plan to consider the $d$HSIC in this setting.

## B  Proof of independence in representation

*Proof.* Without loss of generality, we can write $\mathcal{I}$ as independence between two variables $Z_i \perp\!\!\!\perp Z_j$ for some $(i, j)$. Under infinite data, the empirical distribution $p_{\mathrm{data}}(X, S)$ is close to $p(X, S)$, the real distribution:

$$
\begin{aligned}
\hat{q}_\phi(Z_i, Z_j) &= \int q_\phi(Z_i, Z_j \mid X, S) p_{\mathrm{data}}(X, S) \\
&= \int p_\theta(Z_i, Z_j \mid X, S) p_{\mathrm{data}}(X, S) \\
&= \int p_\theta(Z_i, Z_j \mid X, S) p(X, S) \\
&= p(Z_i, Z_j) \\
&= p(Z_i)(Z_j)
\end{aligned}
$$

.                                                                                                                        $\square$

## C  Linear Gaussian system for independence

Let $(n, m, k, d) \in \mathbb{N}^4$, $A = [A_1, ..., A_n]$, $B = [B_1, ..., B_m]$, $C = [C_1, ..., C_k]$, $\lambda \in \mathbb{R}^+$. We choose our linear system with random matrices:

$$
\begin{aligned}
\forall j \le n, A_j &\sim \mathrm{Normal}(0, \frac{I_d}{n}) \\
\forall j \le m, B_j &\sim \mathrm{Normal}(0, \frac{I_d}{m}) \\
\forall j \le k, C_j &\sim \mathrm{Normal}(0, \frac{I_d}{k}).
\end{aligned}
\tag{12}
$$

Having drawn these parameters, the generative model is:

$$
\begin{aligned}
v &\sim \mathrm{Normal}(0, I_n) \\
u &\sim \mathrm{Normal}(0, I_m) \\
x|u, v &\sim \mathrm{Normal}(Av + Bu, \lambda I_d + CC^T).
\end{aligned}
\tag{13}
$$

The marginal log-likelihood $p(x)$ is tractable:

$$x \sim \text{Normal}(0, \lambda I_d + CC^T + AA^T + BB^T). \tag{14}$$

The complete posterior $p(u, v \mid x)$ is tractable:

$$\begin{aligned}
\Sigma^{-1} &= I_{n+m} + [A, B]^T (\lambda I_d + CC^T)^{-1}[A, B] \\
H_\mu &= \Sigma[A, B]^T (\lambda I_d + CC^T)^{-1} \\
u, v \mid x &\sim \text{Normal}(H_\mu x, \Sigma).
\end{aligned} \tag{15}$$

## D Proof of equivalence between HSIC and MMD

*Proof.* The proof relies on sum manipulations. First, we carefully write the case where $s$ is binary without loss of generality. Let us assume $M$ samples from the joint $(x, s)$ and let us reorder them such that $s_0 = ... = s_N = 0$ and $s_{N+1} = ... = s_M = 1$. In that case,

$$HSIC = \frac{1}{M^2} \sum_{ij}^{M} k_{ij} l_{ij} + \frac{1}{M^4} \sum_{ijkl}^{M} k_{ij} l_{kl} - \frac{2}{M^3} \sum_{ijk}^{M} k_{ij} l_{ik}$$

$$HSIC = \frac{1}{M^2} \sum_{i=0}^{N} \sum_{j=0}^{N} k_{ij} + \sum_{i=N+1}^{M} \sum_{j=N+1}^{M} k_{ij} + \frac{N^2 + (M-N+1)^2}{M^4} \sum_{ij}^{M} k_{ij}$$
$$- \frac{2}{M^3} \left( N \sum_{i=0}^{N} \sum_{j}^{M} k_{ij} + (M-N+1) \sum_{i=N+1}^{M} \sum_{j}^{M} k_{ij} \right)$$

$$HSIC = \frac{(M-N+1)^2}{M^4} \sum_{i=0, j=0}^{N} k_{ij} + \frac{N^2}{M^4} \sum_{i=N+1, j=N+1}^{M} k_{ij} - 2 \frac{N(M-N+1)}{M^4} \sum_{i=0}^{N} \sum_{j=N+1}^{M} k_{ij}$$

$$HSIC = \frac{N^2(M-N+1)^2}{M^4} \left( \frac{1}{N^2} \sum_{i=0, j=0}^{N} k_{ij} + \frac{1}{(M-N+1)^2} \sum_{i=N+1, j=N+1}^{M} k_{ij} \right.$$
$$\left. -2 \frac{1}{N(M-N+1)} \sum_{i=0}^{N} \sum_{j=N+1}^{M} k_{ij} \right).$$

Above, the term inside the parenthesis is the V-statistic for the MMD between $q(z \mid s = 0)$ and $q(z \mid s = 1)$. In the general case of $s$ discrete, we then have a sum of MMD weighted by the values of the empirical $p(s)$. □

## E Nuisance factors presentation for scRNA-seq

- $s^1$: proportion of transcripts which confidently mapped to a gene;
- $s^2$: proportion of transcripts mapping to the genome, but not to a gene;
- $s^3$: proportion of transcripts which did not align;
- $s^4$: proportion of transcripts whose UMI sequence was corrected by the alignment procedure;
- $s^5$: proportion of transcripts whose barcode sequence was corrected by the alignment procedure.

## F Graphical model for scRNA-seq

Our probabilistic graphical model is a modification of the single-cell Variational Inference model [33]. The main difference is the addition of latent variable $u$ and the node for the sequencing errors $s$.

Figure 5: Our proposed modification of the scVI graphical model. Shaded vertices represent observed random variables. Empty vertices represent latent random variables. Shaded diamonds represent constants, set a priori. Empty diamonds represent global variables shared across all genes and cells. Edges signify conditional dependency. Rectangles ("plates") represent independent replication.

**Generative model** Let $\ell_\mu, \ell_\sigma \in \mathbb{R}_+^B$ and $\theta \in \mathbb{R}_+^G$. Let $f_w$ (resp. $f_h$, $f_{\mu_s}$ and $f_{\sigma_s}$) be a neural network with exponential (resp. sigmoid, sigmoid and exponential) link function. Each datapoint $(x_n, s_n)$ is generated according to the following model. First, we draw the latent variables we wish to perform inference over:

$$z_n \sim \text{Normal}(0, I) \tag{16}$$
$$u_n \sim \text{Normal}(0, I) \tag{17}$$
$$\ell_n \sim \text{LogNormal}(\ell_\mu, \ell_\sigma^2) \tag{18}$$
$$\tag{19}$$

$z$ will encode the biological information, $u$ the technical information from the alignment process and $l$ the sampling intensity. Then, we have some intermediate hidden variables useful for testing and model interpretation that we will integrate out for inference:

$$w_{ng} \sim \text{Gamma}(f_w^g(z_n, u_n), \theta) \tag{20}$$
$$y_{ng} \sim \text{Poisson}(\ell_n w_{ng}) \tag{21}$$
$$h_{ng} \sim \text{Bernoulli}(f_h^g(u_n)) \tag{22}$$
$$\tag{23}$$

Physically, $w_{ng}$ represents the average proportion of transcripts aligned with gene $g$ in cell $n$. $y_{ng}$ represents one outcome of the sampling process. $h_{ng}$ represents some additional control for the zeros that come from alignment. Finally, the observations fall from:

$$x_{ng} = \begin{cases} y_{ng} & \text{if } h_{ng} = 0, \\ 0 & \text{otherwise.} \end{cases} \tag{24}$$
$$s_{nj} \sim \text{Normal}(f_{\mu_s}^j(u_n), f_{\sigma_s}^j(u_n)) \tag{25}$$

where we constrained the mean $f_{\mu_s}^j(u_n)$ to be in the 0-1 range since $s$ is a vector of individual proportions. This model is a very competitive solution for representing single-cell RNA sequencing data.

**Variational approximation to the posterior** Applying AEVB with $X = \{x, s\}, Z = \{z, l, u, w, y, h\}$ seems doomed to fail since some of the variables are discrete. Fortunately, variables

$\{w, y, h\}$ can be integrated out analytically: $p(x \mid l, z, u)$ is a Zero-Inflated Negative Binomial distribution. We then apply AEVB with $X = \{x, s\}, Z = \{z, l, u\}$ with the mean-field variational posterior:

$$q(z, l, u \mid x, s) = q(z \mid x)q(l \mid x)q(u \mid x, s).$$

The evidence lower bound is

$$\log p_\theta(x, s) \geq \mathbb{E}_{q_\phi(z|x)q_\phi(l|x)q_\phi(u|x,s)} \log p_\theta(x \mid l, z, u) + \mathbb{E}_{q_\phi(u|x,s)} \log p_\theta(s \mid u)$$
$$- D_{KL}(q_\phi(z \mid x) \| p(z)) - D_{KL}(q_\phi(l \mid x) \| p(l)) - D_{KL}(q_\phi(u \mid x, s) \| p(u)) \tag{26}$$

When using the reparametrization trick [1], all the resulting quantities can be analytically derived and differentiated. Parameters $\ell_\mu, \ell_\sigma^2$ are set to the mean and average of the number of molecules in all cells of the data (in log-scale). Parameters $\theta$ are learned with variational Bayes, and treated as global variables in the training procedure.

**Bayesian hypothesis testing**   We can capitalize on our careful modeling to query the Bayesian model. One particular flavor of Bayesian Hypothesis Testing is called Differential Expression in the biostatistics literature. Given two sets of samples $\{x_a \mid a \in A\}$ and $\{x_b \mid b \in B\}$, we would like to test whether a particular gene $g$ is more expressed in population $A$ or in population $B$.

More formally, for each gene $g$ and a pair of cells $(z_a, u_a), (z_b, u_b)$ with observed gene expression $(x_a, x_b)$ and quality control metrics $(s_a, s_b)$, we can formulate two models of the world under which one of the following hypotheses is true:

$$\mathcal{H}_1^g := \mathbb{E}f_w^g(z_a, u) > \mathbb{E}f_w^g(z_b, u) \quad \text{vs.} \quad \mathcal{H}_2^g := \mathbb{E}f_w^g(z_a, u) \leq \mathbb{E}f_w^g(z_b, u)$$

Where the expectation is taken over $u$ to integrate out the technical variation. Evaluating the likelihood ratio test for whether our datapoints $(x_a, x_b), (s_a, s_b)$ are more probable under the first hypothesis is equivalent to writing a Bayes factor:

$$K = \log_e \frac{p(\mathcal{H}_1^g \mid x_a, x_b)}{p(\mathcal{H}_2^g \mid x_a, x_b)}$$

where the posterior of these models can be approximated via the variational distribution:

$$p(\mathcal{H}_1^g \mid x_a, x_b) \approx \iint_{z_a, z_b, u_a, u_b} p(f_w^g(z_a, u_a) \leq f_w^g(z_b, u_b))dq(z_a \mid x_a)dq(z_b \mid x_b)dp(u_a)dp(u_b).$$

We can use Monte Carlo integration to compute these integrals because all the measures have low dimension.