[Reviews · NeurIPS 2018]

Reviewer 1



In this work, the authors propose an approach to enforce independence among components of latent representations learned with AEVB. This amounts to adding a penalty term based on the Hilbert-Schmidt Independence Criterion to the ELBO. A set of experiments suggest the approach learns independent components. The problem is well-motivated, and theoretic justification is given for why the approach could be expected to work. The experiments are also reasonable, but the lack of interpretation of the results on the real datasets makes it difficult to tell whether the results are in some sense “good.” There is also little theoretical comparison between the proposed approach and similar (cited) work. === Major comments As mentioned above, the lack of explanation of the experimental results makes it difficult to interpret them. For example, in Table 1, the *worst* results for the lighting group and direction are in bold. Presumably, that is actually good because it means the latent representations are ignoring the nuisance (lighting) information. However, a more clear explanation would make this point more obvious. Similarly, it is not obvious what the evaluation metrics used on the sc-seq data really show. The main difference compared to \beta-TCVAE and VFAE seems to be the choice of mutual information measure. Thus, it would be helpful to more explicitly compare dHSIC and the measures used in the other approaches. Otherwise, it is not clear why we would expect the proposed approach to outperform those. === Minor comments At the end of Section 3.1, the authors propound that “the independence properties… will often not be respected by the approximate posterior.” While I agree with this, a reference or two which either prove or demonstrate this empirically would be helpful. Figure 2 should also show results for an unconstrained VAE as a baseline. For reference, Table 1 should include unconstrained VAE results. This would make more clear what is “lost” (in terms of accuracy) by encouraging independence among the latent features. === Typos, etc. “equation 5” -> “Equation 5” The “superscript” notation (presumably indicating a single dimension/variable in the respective space) is not defined. The fonts on Figure 2 should be increased. Also, it is not understandable when printed in black and white. Different marker types could fix that. “has be shown” -> “has been shown” MMD is not defined. “Dirichlet kernel” in Proposition 2. This seems more like the definition of the Dirac delta function. The references are inconsistently formatted. Also, some of the names are misspelled, like “Schlkopf”. --- After author feedback The authors have addressed my concerns.

Reviewer 2



The authors propose a new method to enforce independence in the approximate posterior learned via auto-encoding variational Bayes. Specifically, this is achieved by maximizing a new lower bound that is obtained as the sum of the standard ELBO and a penalization term. The penalization term enforces independence through a kernel-based measure of independence, the Hilbert-Schmidt Information Criterion (HSIC). The authors showcase the advantages of their method over state-of-the-art alternatives in three applications: (i) learning an interpretable latent representation in a case study for which exact posterior inference is tractable; (ii) learning a representation of a face dataset (the extended Yale B dataset) that is invariant to lighting; (iii) denoising of single-cell RNA sequencing data. The paper is well written, the presented experiments are convincing, and the method has the potential of being used to tackle problems in computational biology and other fields. I have only (very) minor comments: 1) the authors could motivate their work a bit more in the introduction, maybe by extending on the concept introduced in section 3: “Independence between certain components of the representation aids in interpretability.”; 2) the authors could add some sentence to give the intuition of the background/methods before proceeding to a formal description; While this is not crucial, I think it would make the paper accessible to a larger audience, which could be important to encourage the use of the method. ------------------------- The authors have addressed my (very) minor comments in their response.

Reviewer 3



This paper proposes to regularize AEVB objective with kernel based measures of independence (d-variable Hilbert Schmidt Independence Criterion) to learn latent representations with desired independence properties. Experiments are performed to use this kind of regularization to test its effectiveness on 3 scenarios: learning interpretable representations, learning invariant representation and denoising representations. The motivation behind the proposed approach is clear and seems to yield desirable representations. The main question to the authors is that for purposes of stochastic optimization, is the empirical estimate of HSIC computed over each mini-batch separately? And how much of a slowdown do you observe practically while optimizing with the regularized objective?